# Caffeine Intake throughout Pregnancy, and Factors Associated with Non-Compliance with Recommendations: A Cohort Study

**DOI:** 10.3390/nu14245384

**Published:** 2022-12-18

**Authors:** María Rosario Román-Gálvez, Sandra Martín-Peláez, Loreto Hernández-Martínez, Naomi Cano-Ibáñez, Rocío Olmedo-Requena, Juan Miguel Martínez-Galiano, Aurora Bueno-Cavanillas, Carmen Amezcua-Prieto

**Affiliations:** 1Unit of Clinical Management Alhama de Granada, Andalusian Health Service, 18120 Alhama de Granada, Spain; 2Department of Nursing, University of Granada, 18006 Granada, Spain; 3Department of Preventive Medicine and Public Health, Faculty of Medicine, University of Granada, 18016 Granada, Spain; 4Instituto de Investigación Biosanitaria (ibs.Granada), 18014 Granada, Spain; 5Obstetrics and Gynaecology Service, Hospital Clínico San Cecilio, 18016 Granada, Spain; 6Consortium for Biomedical Research in Epidemiology and Public Health (CIBERESP), 28029 Madrid, Spain; 7Department of Nursing, University of Jaén, 23071 Jaén, Spain

**Keywords:** caffeine, recommendations, lifestyles, pregnancy, compliance

## Abstract

Maternal caffeine consumption is associated with adverse gestational outcomes. The aim of this study was to assess the intake of caffeine and factors associated with the non-adherence to caffeine intake recommendations in a cohort of 463 women before (T0) and in each trimester of gestation (T1, T2, and T3), by using validated questionnaires. Caffeine intake (median (mg/day), IQR) was 100.0 (181.1) at T0, 9.42 (66.2) at T1, 12.5 (65.6) at T2, and 14.0 (61.1) at T3 (*p* < 0.001). Non-compliance prevalence (intake > 200 mg/day) was 6.2% at T1, 4.2% at T2, and 2.7% at T3. Not being an active smoker at T1 (OR = 0.17; 95% CI 0.05–0.59) and T2 (OR = 0.22; 95% CI 0.09–0.52), adherence to the Mediterranean Diet at T1 (OR = 0.50; 95% CI 0.28–0.88) and T2 (OR = 0.39; 95% CI 0.15–1.02), and moderate physical activity at T1 (OR = 0.50; 95% CI 0.28–0.88) were inversely associated with caffeine consumption. Although caffeine intake may be considered low, intake prevalence increases throughout pregnancy. Although the main source of caffeine during pregnancy is coffee, attention must be also paid to the increasingly intake of chocolate, of which the effect during pregnancy is controversial. Smoking, non-adherence to a good quality diet, and light physical activity are associated with a higher caffeine intake and a lower compliance with caffeine intake recommendations. Perinatal dietary and lifestyle educational policies are needed.

## 1. Introduction

Caffeine is a psychoactive substance, an alkaloid from the methylxanthines family, which freely crosses the placental barrier [1]. In the maternal liver, cytochrome P450 1A2 (CYP1A2) enzyme metabolizes 90% of caffeine. This enzyme is absent from the fetal liver and placenta [1,2,3]. The average half-life of caffeine increases in pregnancy, reaching 9–11 h in the third trimester [2]. Main sources of dietary caffeine include coffee, tea, and chocolate. Additionally, caffeine is added to cola products and energy drinks [4].

During pregnancy, the maximum limit of caffeine intake recommended is 200 mg/day (about two cups of coffee) [5,6]. This is because maternal caffeine consumption has been associated with miscarriage, stillbirth, low birth weight or small for gestational age, and with overweight or obese offspring [7,8]. A meta-analysis showed that the risk of pregnancy loss increases by 19% for each increase of 150 mg/day of caffeine, and 8% for each increase of two cups of coffee/day [9]. Another meta-analysis found that 100 mg of caffeine per day during pregnancy was also associated with 14 and 26% increased risk of pregnancy loss in cohort and case-control studies, respectively [10]. Additionally, a higher risk of bleeding in early pregnancy was identified amongst pregnant women who ingested two or more cups of coffee per day before pregnancy [11].

Knowledge about the adherence to the caffeine intake recommendations during pregnancy is rather low [12,13]. In addition, little is known about lifestyles associated with caffeine consumption. Pre-pregnancy smoking has been associated with non-compliance with caffeine intake recommendations during pregnancy [11,13,14]. Few studies have associated prenatal caffeinated beverage intake with the quality of diet [15]. The association of other lifestyles, such as physical activity or insomnia, with caffeine intake has not been explored. Thus, in order to promote healthy lifestyles according to caffeine intake during pregnancy, studies revealing the habits of caffeine intake in women of childbearing age, assessed not only in the first trimester of pregnancy, are needed [16]. The aim of this study was to evaluate the intake of caffeine before and throughout pregnancy, variations in caffeine consumption, and factors associated with the non-adherence to caffeine intake recommendations in pregnancy.

## 2. Materials and Methods

### 2.1. Study Design

The reporting of this work is compliant with the STrengthening the Reporting of OBservational studies in Epidemiology (STROBE) guidelines. We carried out a prospective cohort study (PROY-PP 2015-01): ‘Lifestyles in pregnant women’ in which other habits besides caffeine intake were measured: smoking [17], insomnia [18], physical activity [19], and Adherence to the Mediterranean Diet (AMD). Volunteer midwives selected women from February 2013 to February 2016, from 47 Health Centers. The sample size necessary to identify a difference in non-compliance with caffeine intake greater than or equal to 10% between different time periods during pregnancy, considering an alpha risk of 0.05 and a beta risk of 0.10 in a two-sided test and assuming a dropout rate of 30%, was N = 269. A total of 518 pregnant women were recruited and 89.3% of them (N = 463) had some follow-up throughout pregnancy. Fifty-five women were not included because of miscarriage (N = 32) or refusing to participate (N = 24) after the first interview. Before the second and third interview, some participants were lost to follow-up (N = 25 and N = 9, respectively). Figure 1

The reference population included pregnant women managed in health centers run by the Andalusian Health Service from Seville, Huelva, Granada, and Jaen (Spain). They met the following selection criteria: healthy pregnant women, consent to participate in the study, a clinic appointment before 14 weeks of gestation, and without language, cognitive, or understanding barriers. We excluded pregnant women with multiple pregnancies, or with chronic diseases altering diet or physical activity, such as diabetes, gestational diabetes, hypertension, gestational hypertension, heart failure, respiratory problems, moderate or severe renal or hepatic disease, and neurological or musculoskeletal diseases that influence mobility.

Throughout pregnancy, a trained interviewer conducted three structured interviews, either in-person (T1) or over the telephone (T2 and T3). In the first interview, taken at the 12th gestational week (GW) (T1), information about the first trimester of pregnancy and from the three months before getting pregnant (T0) was collected. The second and the third interviews were conducted during routine follow-up appointments at the second trimester (T2, at 24 GW ± 2 weeks), and at the third trimester (T3, at 32 GW), respectively.

Data collected at T1 were as follows: (1) sociodemographic variables: age, nationality, education level, paid work, parity, and social class (I high, II medium-high, III medium, IV medium-low, V low), taking into account the higher social class, either from the woman’s or her partner’s occupation [20] and (2) pre-pregnancy anthropometrics data (weight, height, and BMI). In addition, at T1, T2, and T3 data about smoking and alcohol habits, insomnia (by AIS), categorized as no insomnia (<8) or insomnia (≥8) [21], physical activity (IPAQ-short form) [22], and adherence to Mediterranean Diet (AMD), using the 13-point MEDAS-score, categorized as low adherence (<8) and high AMD (≥8) [23,24] were obtained.

### 2.2. Caffeine Intake Calculations

Caffeine intake was measured at pre-pregnancy (3 months before pregnancy), in T1, T2, and T3. The intake of caffeine was calculated from the consumption of beverages (coffee, tea, cola drinks, energetic drinks) and milk and dark chocolates, through questions present in a validated food questionnaire [25]. Transformation to mg/day of caffeine from each source was calculated according to the literature (Appendix A) [26,27] as follows: 100 mg per cup of caffeinated coffee, 2 mg per cup of decaffeinated coffee, 39 mg per cup of tea, 20 mg per bottle of 200 mL of cola-drinks, 64 mg per 200 mL of energy drink, 6 mg per square of milk chocolate, and 23 mg per square of dark chocolate. Caffeine ingestion recommendation was achieved if caffeine consumption was lower than 200 mg/day [5,6] during pregnancy.

### 2.3. Statistical Analyses

Categorical variables required absolute and relative frequencies and 95% CI, whereas for quantitative variables, median and IQR were calculated. The normality of the continuous variables was assessed by using normal probability plots and the Kolmogorov–Smirnoff test. Kruskal Wallis nonparametric test was used for analyzing non-normally distributed variables.

To assess factors associated with the lack of adherence to caffeine recommendations during each trimester of pregnancy (T1, T2, and T3), multiple logistic regressions were applied (Models 1, 2, and 3). Factors related to caffeine intake were considered in all models as potential confounders and were as follows: age (18–25; 26–30; 31–35; >35), social class (I–II; III; IV–V), active smoker (no, yes), parity (0,1, ≥2), paid work (yes, no), Body Mass Index (normal, overweight, and obesity), physical activity level (light, moderate, and vigorous), AMD (low < 8, ≥8), and insomnia (no < 8, yes ≥ 8). We considered statistically significant a *p* value below 0.05 and a confidence level of 95%. We used STATA 15 College Station, TX, USA statistical packages for the analysis.

## 3. Results

### 3.1. Population Characteristics

The average age of the cohort of pregnant women was 31.2 years (SD 4.87). Most of the participants had Spanish nationality (94%), 42.5% had university studies, and 53.3% were at moderate-high social class (I to III). During pre-pregnancy, 70.6% of the sample had paid work. Around half of pregnant women (51.6%) were primiparous.

Median and IQR caffeine intake before pregnancy was 100.01 mg/day (IQR 181.05), 9.42 mg/day (IQR 66.24) at T1, 12.1 mg/day (IQR 65.57) at T2, and 14.00 mg/day (IQR 61.1) at T3 (*p* < 0.001). Median and IQR were higher in active smokers’ pre-pregnancy and throughout pregnancy (*p* < 0.001). Those women with non-adherence to the Mediterranean Diet, in T1 (*p* < 0.001), T2 (*p* = 0.03), and T3 (*p* < 0.001), had higher mg/day caffeine intake Table 1.

### 3.2. Prevalence and Changes in Caffeine Intake throughout Pregnancy

Prevalence of caffeine intake before and throughout pregnancy is presented in Figure 2. More than 80% of the participants ingested any of the caffeine sources investigated pre-pregnancy or during pregnancy. From those, 27% ingested 200 mg/day or more at pre-pregnancy, decreasing this prevalence throughout pregnancy (6.2%, 4.2%, and 2.7% for T1, T2, and T3, respectively, data not shown). The highest prevalence of intake was found for milk-chocolate, at all timepoints (73.9%, 50.5%, 74.1%, and 76.2% for pre-pregnancy, T1, T2, and T3, respectively). Prevalence of milk-chocolate intake at T2 and T3 surpassed that found at pre-pregnancy. The second highest prevalence was found for cola-drinks at pre-pregnancy (62.6%) and T1 (36.9%), being similar to decaffeinated coffee intake prevalence at T2 and T3 (around 38%). For all the sources investigated, prevalence intake decreased at T1. Compared to T1, the prevalences of consumption of milk chocolate and decaffeinated coffee were the ones that increased the most in T2 and T3.

Table 2 shows the frequencies of women who started to consume and increased or decreased the amount of caffeine intake throughout pregnancy, compared to pre-pregnancy. The 17.0% and the 9.3% of the participants not consuming decaffeinated coffee or cola-drinks at pre-pregnancy, respectively, started consuming these sources at T1. Smaller percentages of participants started consuming the rest of the caffeine sources investigated, reaching the 5.5%, 2.1%, and 5.5% in the case of coffee, tea, and dark chocolate, respectively. At T3, milk chocolate showed the highest and sharper frequency of participants increasing the intake throughout pregnancy compared to pre-pregnancy, reaching 58.9% in those not consuming milk chocolate before pregnancy, and 42.1% in those consuming milk-chocolate before pregnancy. Dark chocolate showed the highest consumption increase in T1 (10.2%), compared to pre-pregnancy consumption. 

### 3.3. Non-Compliance with Recommendations and Factors Associated

Non-compliance with recommendations of caffeine intake before pregnancy (>300 mg/day) was 10.8% (n = 50). During pregnancy, 6.2% of women (n = 28) in T1, 4.2% (n = 19) in T2, and 2.7% (n = 12) in T3 exceed the caffeine intake recommendations (>200 mg/day), Table 3.

Table 3 shows the logistic regression of factors associated with non-compliance with caffeine recommendation throughout pregnancy. During pregnancy (T1 and T2), not being an active smoker was inversely associated with caffeine consumption above recommendations (>200 mg/day), (aOR T1 = 0.17; 95% CI 0.05–0.59; aOR T2 = 0.22; 95% CI 0.09–0.53). Adherence to the Mediterranean Diet in T1 and T2 (aOR T1 = 0.50; 95% CI, 0.28–0.88; aOR T2 = 0.39; 95% CI, 0.15–1.02) and moderate physical activity in T1 (aOR T1 = 0.50; 95% CI, 0.28–0.88) were also inversely associated. Non-paid work at T1 was a risk factor of non-compliance with caffeine intake recommendations in T1 (aOR T1 = 2.09; 95% CI, 1.10–3.98) and it was a protective factor in T3 (aOR T3 = 0.34; 95% CI, 0.12–0.93).

## 4. Discussion

The present study shows that, prevalence of caffeine intake is lower during pregnancy compared to pre-pregnancy, although it increases throughout pregnancy. In addition, mg of caffeine intake decreases throughout pregnancy, below a half of that consumed at pre-pregnancy. At the beginning of pregnancy, 6 out of 100 women were above the caffeine intake recommendations. The greatest contribution to the daily caffeine intake before and during pregnancy arose from the consumption of coffee, cola drinks, milk chocolate, and decaffeinated coffee. Active smokers presented a higher caffeine intake before and during pregnancy. A lower consumption of total caffeine intake per day during the first trimester of pregnancy was observed in those who performed moderate physical activity and had higher adherence to the Mediterranean Diet.

Previous studies conducted in New York [28] and the United Kingdom [29] also used questionnaires to collect caffeine intake—Food Frequency Questionnaire and CAT Questionnaire. In the assessment of caffeine intake, other authors have considered, as we did, cola drinks and tea [12] or energy drinks and chocolate [27,30].

In our study, and in American pregnant women, coffee was the main source of caffeine, contributing over 60% of total consumption [14]. This was also observed in Asian pregnant women. Data from the Korean pregnant outcome study (KPOS) showed that before conception, 17% of women were moderate coffee drinkers (1 cup/day) and 18% were heavy coffee drinkers (>2 cups/day) [11]. It should be noted in our study that, from conception, milk chocolate is consumed more than tea. This was not happening during the preconception stage and differs from the results of other European studies [30,31], where chocolate consumption decreases during pregnancy. In addition, the increased prevalence of consumption of milk-chocolate during pregnancy can be translated into an increase in energy intake (around 535 kcal/100 g), and saturated fats (around 19 g/100 g), which could have consequences on weight control and maintenance of a healthy lipid profile, altered in pregnancy [32,33]. Furthermore, although some studies show a protective effect of chocolate on medical conditions associated to pregnancy, such as preeclampsia [34], recent studies demonstrate harmful effects, such as fetal ductal constriction in animal models [35] and in humans [36]. Therefore, chocolate consumption during pregnancy should be monitored.

Data from different studies reveal a caffeine intake lower than the recommendations in pregnant women; for example, 44 mg/day at 17th GW in Norwegian women [30] or a median value of 58 mg at 22nd GW, also in Norwegian women [37]. In British pregnant women, average caffeine intake during pregnancy (mostly from tea) was 128.6 mg/day [28]. In our study, more than 96% of pregnant women ingested less than 200 mg/day of caffeine. A similar behavior has been observed in the Warsaw region, where only 2% of the respondents exceeded the safe dose of 200 mg [29]. In Australia, 70% of Australian pregnant women seemed to comply [13]. On the contrary, more than 40% of pregnant women in Finland reported consuming more than the recommended maximum of 200 mg of caffeine per day [38].

Other studies [12,28,30,39,40,41], as we did, have shown a decrease in the average consumption of caffeine intake through pregnancy, compared to pre-pregnancy.

We found that smoking during pregnancy was associated with higher caffeine intake. In concordance with our results, previous studies have also associated smoking before pregnancy with higher caffeine consumption during pregnancy [11,13,14]. Smokers may drink more caffeine because smoking increases the rate of caffeine metabolism (upregulates CYP1A2), which means that smokers may need to drink more caffeine to experience the same effects as non-smokers. Additionally, prenatal caffeinated beverages, not including coffee, such as soft drinks/soda/pop/sugary drinks/fizzy drinks and energy drinks, have been related to non-AMD [15]. In our study, we found a higher caffeine intake during the first trimester of pregnancy in those women with non-AMD. Furthermore, we found that moderate physical activity in the first trimester of pregnancy seems to be associated with lower mg/day of caffeine intake. In a recent meta-analysis [42], it has been found that although pregnant women with intended pregnancies reported healthier eating and diet quality, lower caffeine intake, and higher levels of physical activity than women with unintended pregnancies, at preconception, most of the women are not engaging in diet and physical activity behaviors that are known to optimize perinatal outcomes, independently of their intention to get pregnant. In this sense, and considering our results, there is a need to provide women with strategies to adopt healthy lifestyle behaviors, in particular, regarding diet and physical activity during the preconception and antenatal periods.

### 4.1. Study Limitations

There may be a potential risk of memory bias because caffeine intake before pregnancy was established when the woman was already pregnant. Medications and dietary supplements were not considered in the caffeine measurement. We considered exclusively beverages and foods containing caffeine. Caffeine intake in our sample was rather low, only 6.2% of women exceeded the dose of 200 mg at T1, decreasing throughout pregnancy.

### 4.2. Study Strengths

This is a prospective cohort study where selection of health centers should represent healthy pregnant women from rural and urban settings. It also allows a comparison of multiple sources of caffeine consumption between different trimesters of pregnancy and the pre-gestational stage. The same researcher collected the data, giving uniform information and consistency to recorded data. We used questions from a validated food frequency questionnaire (FFQ) to collect caffeine intake. The FFQ was previously validated [25] and has been used previously to measure caffeine in pregnant women [39]. Women who dropped out from the study were similar in demographic variables from those who finished the study.

Accumulating scientific evidence advises pregnant women and women contemplating pregnancy to avoid caffeine [8]. Health information about adequate caffeine intake should focus on all pregnant women. Because there seems to be a general aggrupation of unhealthy lifestyles during pregnancy, health educators must be alert and pay special attention to women who add unhealthy lifestyles from before conception, such as active smokers, light physical activity women, and those with poor quality diets. Future research should be focused on the influence of insomnia, depression, or anxiety on caffeine consumption in pregnancy, as well as the caffeine intake patterns of pregnant women’s partners and their influence on them.

## 5. Conclusions

Caffeine intake by the study group of pregnant women may be considered low, although intake prevalence increases throughout pregnancy. Although coffee is the main source of caffeine during pregnancy, consumption of milk-chocolate increases during pregnancy, which could have consequences regarding weight control and lipid profile. Pregnant women are mostly adhering to current caffeine intake guidelines. Higher caffeine intake at pregnancy is associated with other unhealthy habits during pregnancy, such as smoking, light physical activity, and non-adherence to the Mediterranean Diet.

## Figures and Tables

**Figure 1 nutrients-14-05384-f001:**
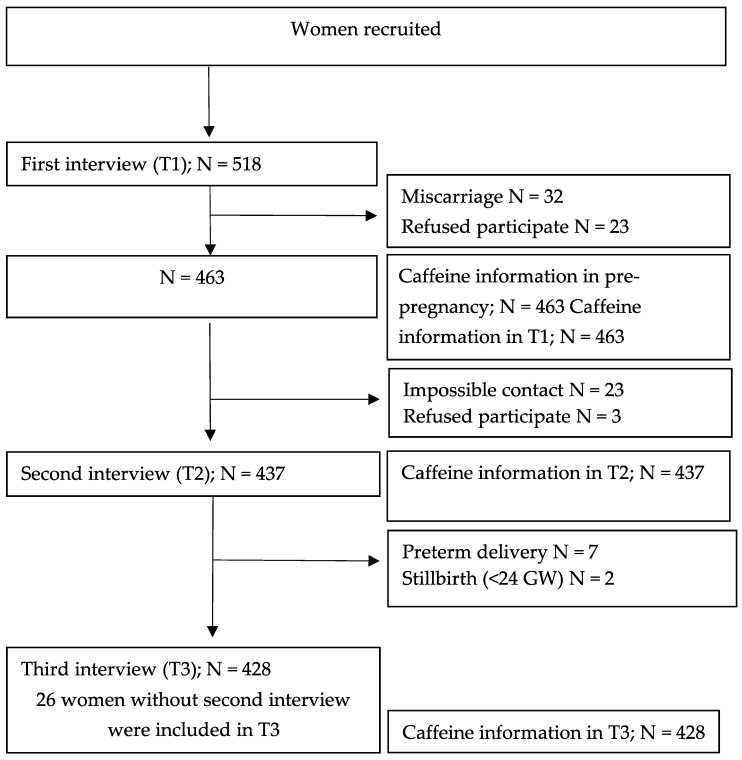
Cohort flow chart.

**Figure 2 nutrients-14-05384-f002:**
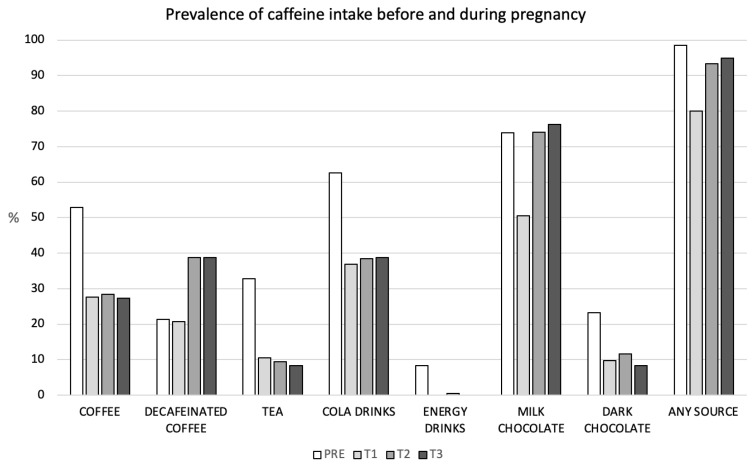
Prevalence of caffeine intake before and during pregnancy. T0 (pre-pregnancy), T1, T2, T3: first (12th gestational week), second (24th ± 2 gestational weeks), and third (at 32nd gestational week) trimester of gestation, respectively.

**Table 1 nutrients-14-05384-t001:** Median and interquartile range of caffeine intake before and during pregnancy.

	N = 463 (Pre-Pregnancy)		N = 463 (T1)		N = 437 (T2)		N = 428 (T3)	
mg/day Caffeine Median (IQR)	100.01 (181.05)		9.42 (66.24)		12.1 (65.57)		14.0 (61.1)	
Age	n	median (IQR)	n	median (IQR)	n	median (IQR)	n	median (IQR)
18–25	53	100 (128.18)	53	13.28 (49.85)	49	18.71 (40.55)	49	20.14 (36.88)
26–30	137	75.85 (154.85)	137	6.85 (37.41)	132	12 (28.57)	125	12.72 (40.42)
31–35	185	100.22 (194.26)	185	8.49 (59.97)	171	10.22 (66)	169	12 (62.28)
>35	88	105.59 (182.73)	88	19.42 (99.56)	85	29.42 (96.95)	85	19.28 (96.48)
	*p* = 0.83		*p* = 0.12		*p* = 0.06		*p* = 0.78	
Social Class	n	median (IQR)	n	median (IQR)	n	median (IQR)	n	median (IQR)
I–II	157	102 (184.34)	157	6.28 (99.93)	152	13.29 (96.28)	149	14.5 (84.14)
III	90	90 (187.86)	90	11.37 (41.43)	86	10.81 (74.57)	83	12 (96.71)
IV–V	216	91.85 (170.48)	216	10.45 (55.91)	199	12.28 (47.42)	196	14.17 (33.73)
	*p* = 0.90		*p* = 0.77		*p* = 0.65		*p* = 0.63	
Paid work	n	median (IQR)	n	median (IQR)	n	median (IQR)	n	median (IQR)
Yes	327	102.57 (184.57)	327	9.42 (99.78)	306	12.28 (96.57)	302	13.42 (45.85)
No	136	69.43 (115.07)	136	9.42 (45.75)	131	12.14 (41.57)	126	14 (30.57)
	*p* = 0.23		*p* = 0.89		*p* = 0.48		*p* = 0.17	
Parity	n	median (IQR)	n	median (IQR)	n	median (IQR)	n	median (IQR)
0	239	102.57 (172.97)	239	6.28 (59.86)	227	12 (55.14)	221	12.85 (45.14)
1	197	68.72 (187.57)	197	11.14 (48)	185	13.14 (56.77)	182	14.64 (69.85)
>1	27	115.67 (132.71)	27	57 (97.96)	25	48 (96.2)	25	7.42 (46)
	*p* = 0.45		*p* = 0.07		*p* = 0.20		*p* = 0.46	
Active smokers	n	median (IQR)	n	median (IQR)	n	median (IQR)	n	median (IQR)
Yes	167	121.42 (215.42)	398	6.38 (49.50)	54	42.53 (104.93)	51	33.2 (103.42)
No	245	66.4 (126.16)	65	39.79 (98.52)	383	11.83 (47.42)	377	12.06 (45.42)
	*p* < 0.001		*p* < 0.001		*p* < 0.001		*p* < 0.001	
AMD	n	median (IQR)	n	median (IQR)	n	median (IQR)	n	median (IQR)
Yes	218	100.01 (215.42)	275	6.28 (49.53)	263	10.22 (58.6)	258	12 (40.28)
No	245	100 (176.71)	188	14.44 (98)	173	16.14 (61.68)	166	24.28 (70.42)
	*p* = 0.32		*p* < 0.001		*p* = 0.03		*p* < 0.001	
BMI	n	median (IQR)	n	median (IQR)	n	median (IQR)	n	median (IQR)
Normal	287	100 (182.58)	287	7.31 (56.76)	273	12.71 (59.95)	268	12.79 (56.85)
Overweight	128	80.79 (183.08)	128	14.24 (98.26)	118	13.65 (59.95)	116	15.75 (71.55)
Obesity	45	119.28 (166.79)	45	10.62 (58.90)	44	6.65 (23.92)	41	16.71 (51.14)
	*p* = 0.39		*p* = 0.18		*p* = 0.22		*p* = 0.91	

*p* value: Kruskal Wallis test; BMI, Body Mass Index; mg, milligrams; AMD: Adherence to the Mediterranean Diet.

**Table 2 nutrients-14-05384-t002:** Frequencies of women starting, increasing, or decreasing the amount of caffeine intake throughout pregnancy, compared to pre-pregnancy.

Compared to Pre-Pregnancy Values	T1	T2	T3
	n = 463	n = 437	n = 428
Coffee			
No consumers	n = 218	n = 207	n = 201
Start consuming	3.7%	3.4%	5.5%
Consumers	n = 245	n = 230	n = 227
Increase intake	1.2%	0.8%	1.8%
Decrease intake	75.5%	80.0%	83.7%
Decaffeinated coffee			
No consumers	n = 364	n = 342	n = 335
Start consuming	17.0%	30.4%	31.0%
Consumers	n = 99	n = 95	n = 93
Increase intake	0.0%	24.2%	24.7%
Decrease intake	69.7%	53.7%	50.5%
Tea			
No consumers	n = 311	n = 291	n = 286
Start consuming	0.3%	3.8%	2.1%
Consumers	n = 152	n = 146	n = 142
Increase intake	1.3%	5.5%	5.6%
Decrease intake	76.3%	88.4%	90.9%
Cola drinks			
No consumers	n = 173	n = 162	n = 163
Start consuming	9.3%	18.5%	22.1%
Consumers	n = 290	n = 275	n = 265
Increase intake	1.7%	13.8%	14.3%
Decrease intake	71.0%	77.1%	77.7%
Energy drinks			
No consumers	n = 424	n = 400	n = 390
Start consuming	0.0%	0.0%	0.00%
Consumers	n = 39	n = 37	n = 38
Increase intake	0.0%	2.7%	0.0%
Decrease intake	100%	97.3%	100%
Milk Chocolate			
No consumers	n = 121	n = 114	n = 112
Start consuming	4.1%	53.5%	58.9%
Consumers	n = 342	n = 323	n = 316
Increase intake	5.60%	40.6%	42.1%
Decrease intake	44.2%	48.0%	45.6%
Dark Chocolate			
No consumers	n = 355	n = 335	n = 330
Start consuming	0.5%	5.7%	5.5%
Consumers	n = 108	n = 102	n = 98
Increase intake	10.2%	16.7%	12.2%
Decrease intake	65.7%	77.5%	86.7%
All sources			
No consumers	n = 7	n = 6	n = 7
Start consuming	28.6%	83.3%	100%
Consumers	n = 456	n = 431	n = 421
Increase intake	5.3%	16.5%	18.1%
Decrease intake	84%	82.8%	80.8%

T1, T2, T3: first (12th gestational week), second (24th ± 2 gestational weeks), and third (at 32nd gestational week) trimester of gestation, respectively.

**Table 3 nutrients-14-05384-t003:** Logistic regression of factors associated with non-compliance with caffeine intake recommendations throughout pregnancy.

	T1 (>200 mg/day)				T2 (>200 mg/day)				T3 (>200 mg/day)			
	Model 1				Model 2				Model 3			
	cOR ^a^	CI 95%	aOR ^b^	CI 95%	cOR	CI 95%	aOR	CI 95%	cOR	CI 95%	aOR	CI 95%
Age												
18–25	1 (Ref.)		1 (Ref.)		1 (Ref.)		1 (Ref.)		1 (Ref.)		1 (Ref.)	
26–30	0.49	0.19–1.27	0.62	0.23–1.67	0.39	0.08–1.79	0.59	0.11–2.99	1.09	0.27–4.43	1.07	0.22–5.02
31–35	0.39	0.16–0.99	0.58	0.22–1.54	0.76	0.15–3.66	1.65	0.29–9.50	2.13	0.49–9.28	1.81	0.35–9.30
>35	0.73	0.26–2.07	0.98	0.33–2.94	0.68	0.12–3.64	1.75	0.25–11.91	0.72	0.17–2.94	0.55	0.11–2.68
Social Class												
I–II	1 (Ref.)		1 (Ref.)		1 (Ref.)		1 (Ref.)		1 (Ref.)		1 (Ref.)	
III	1.07	0.58–2.00	0.95	0.50–1.82	0.76	0.29–1.97	0.95	0.26–3.48	1.99	0.40–9.83	1.74	0.33–9.12
IV–V	1.59	0.95–2.67	1.03	0.55–1.92	1.47	0.60–3.56	1.4	0.46–4.30	0.69	0.27–1.78	0.93	0.28–3.09
Parity												
0	1 (Ref.)		1 (Ref.)		1 (Ref.)		1 (Ref.)		1 (Ref.)		1 (Ref.)	
1	1.52	0.94–2.45	1.41	0.63–3.17	1.49	0.82–2.70	1.21	0.50–2.96	1.03	0.39–2.66	1	0.35–2.86
≥ 2	2.44	0.71–8.43	0.93	0.20–4.28	0.75	0.16–3.37	0.47	0.09–2.48	0.24	0.07–0.86	0.33	0.07–1.53
Paid Work												
Yes	1 (Ref.)		1 (Ref.)		1 (Ref.)		1 (Ref.)		1 (Ref.)		1 (Ref.)	
No	1.76	1.02–3.06	2.09	1.10–3.98	1.13	0.48–2.62	1.16	0.46–2.90	0.32	0.13–0.77	0.34	0.12–0.93
AMD												
Low (<8)	1 (Ref.)		1 (Ref.)		1 (Ref.)		1 (Ref.)		1 (Ref.)		1 (Ref.)	
High (≥8)	0.47	0.28–0.78	0.5	0.28–0.88	0.37	0.15–0.94	0.39	0.15–1.02	1.43	0.59–3.46	1.36	0.52–3.55
Active smokers												
Yes	1 (Ref.)		1 (Ref.)		1 (Ref.)		1 (Ref.)		1 (Ref.)		1 (Ref.)	
No	0.16	0.05–0.54	0.17	0.05–0.59	0.5	0.11–2.19	0.22	0.09–0.53	0.33	0.04–2.57	0.32	0.04–2.58
BMI												
Normal	1 (Ref.)		1 (Ref.)		1 (Ref.)		1 (Ref.)		1 (Ref.)		1 (Ref.)	
Overweight	1.13	0.67–1.90	1.11	0.59–2.12	0.91	0.38–2.17	0.56	0.22–1.46	0.66	0.25–1.76	0.6	0.21–1.69
Obesity	2.2	0.83–5.82	1.47	0.60–3.55	0.66	0.21–2.07	0.43	0.13–1.47	0.54	0.14–2.03	0.64	0.15–2.64
IPAQ												
Light	1 (Ref.)		1 (Ref.)		1 (Ref.)		1 (Ref.)		1 (Ref.)		1 (Ref.)	
Moderate	0.63	0.39–1.02	0.63	0.39–1.02	0.57	0.26–1.25	0.67	0.30–1.49	0.95	0.40–2.24	0.76	0.29–1.95
Vigorous	0.9	0.13–2.70	0.8	0.24–2.64	-	-	-	-	-	-	-	-
Insomnia												
No	1 (Ref.)		1 (Ref.)		1 (Ref.)		1 (Ref.)		1 (Ref.)		1 (Ref.)	
Yes	1.01	0.961.07	1	0.93–1.06	0.93	0.87–1.01	1.02	0.95–1.12	0.92	0.85–1.00	0.93	0.85–1.02

T1, T2, T3: first (12th gestational week), second (24th ± 2 gestational weeks), and third (at 32nd gestational week) trimester of gestation, respectively; ^a^ cOR: Crude odds ratio; ^b^ OR: adjusted odds ratio: adjusted for age, social class, parity, paid work; Adherence to the Mediterranean Diet in T1 (model 1), T2 (model 2), or T3 (model 3); active smokers in T1 (model 1), T2 (model 2), or T3 (model 3); BMI, IPAQ in T1 (model 1), T2 (model 2), or T3 (model 3); and Insomnia in T1 (Model 1), T2 (Model 2), or T3 (Model 3). Abbreviations: AMD, Adherence to the Mediterranean Diet; BMI, Body Mass Index; IPAQ, International Physical Activity Questionnaire; Ref., reference gategory.

## Data Availability

Not applicable.

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
