# Peer review of "Caffeine Intake throughout Pregnancy, and Factors Associated with Non-Compliance with Recommendations: A Cohort Study"

_nutrients, 2022, doi:10.3390/nu14245384_

Round 1

Reviewer 1 Report

Authors discuss and describe the caffeine consumption in pregnancy and compare it with pre-pregnancy consumption. The methods used are described well. The authors also make sure they are covering all the sources of caffeine, and not just coffee. The comparison in the three trimesters is useful and is required for the policies and education of women in different populations.

Some minor revisions are needed as shown below:

Abstract: Rephrase the sentence “Coffee is the main source of caffeine during pregnancy, although consumption of chocolate increases, which effect during pregnancy is controversial.”

study design

Line 10:  Before second and third interview, n= 25 and n=9 women were lost, respectively.” Instead of saying “women were lost”, rephrase it to “lost contact with women” or “ women were difficult to contact”.

Table 1 needs to be fixed. Spread over 2 pages and the top line is “bold”, without mention of what that bold means.

Page 6 of 13, line 1: Change sentence “frequencies of women starting consuming…” to “frequencies of women who started to consume, and increased or decreased the amount of….”

Page 9 of 13, para 3, line 2: change word “assuming” to “contributing”.

Page 9 of 13, para 3, Line 8: Edits to this sentence- “In addition, the increased prevalence of consumption of milk-chocolate during pregnancy can be translated in to an increase in energy intake (around 535 kcal/100 grams), and saturated fats (around 19 grams/100 grams), which could have consequences regarding on weight control and maintenance of a healthy lipid profile, alterated during in pregnancy.”

Reviewer 2 Report

Dear colleagues,

Thank you very much for submitting your manuscript. The evaluation of the intake of caffeine before and throughout pregnancy is essential to study the possible effects on pregnant women. The topic is relevant and exciting to the field of the journal. The text is clear and easy to read. The manuscript has an excellent structure and description. The overall paper is organized and well-written. The literature reviews are insightful and informative.

The figures and the tables are well-presented and easy to read and understand. The presented aspects sufficiently support the conclusions.

I congratulate all the authors for their efforts.

I have only a few remarks to make:

-       The aim of the study is not written in the abstract.

-       I recommend entering several keywords considering the title and the topic addressed. I propose something related to compliance, for example.
